# Mammalian cells internalize bacteriophages and use them as a resource to enhance cellular growth and survival

**Marion C. Bichet**[1,2,3], **Jack Adderley**[4], **Laura Avellaneda-Franco**[1], **Isabelle Magnin-Bougma**[1], **Natasha Torriero-Smith**[1], **Linden J. Gearing**[5,6], **Celine Deffrasnes**[7], **Cassandra David**[7], **Genevieve Pepin**[8], **Michael P. Gantier**[5,6], **Ruby CY Lin**[9], **Ruzeen Patwa**[1], **Gregory W. Moseley**[7], **Christian Doerig**[4], **Jeremy J. Barr**[1] *

1 School of Biological Sciences, Monash University, Clayton, Australia, 2 ACTALIA, Food Safety Department, Saint-Lô, France, 3 University of Lorraine, CNRS, LCPME, Vandœuvre-lès-Nancy, France, 4 School of Health and Biomedical Science, RMIT University, Bundoora, Australia, 5 Centre for Innate Immunity and Infectious Diseases, Hudson Institute of Medical Research, Clayton, Australia, 6 Department of Molecular and Translational Sciences, Monash University, Clayton, Australia, 7 Department of Microbiology, Biomedicine Discovery Institute, Monash University, Clayton, Australia, 8 Medical Biology Department, Université du Québec à Trois-Rivières, Trois-Rivières, Québec, Canada, 9 Centre for Infectious Diseases and Microbiology; The Westmead Institute for Medical Research, Westmead, Australia

* jeremy.barr@monash.edu

**Data Availability Statement:** All relevant data are within the paper and its Supporting Information files.

## Abstract

There is a growing appreciation that the direct interaction between bacteriophages and the mammalian host can facilitate diverse and unexplored symbioses. Yet the impact these bacteriophages may have on mammalian cellular and immunological processes is poorly understood. Here, we applied highly purified phage T4, free from bacterial by-products and endotoxins to mammalian cells and analyzed the cellular responses using luciferase reporter and antibody microarray assays. Phage preparations were applied in vitro to either A549 lung epithelial cells, MDCK-I kidney cells, or primary mouse bone marrow derived macrophages with the phage-free supernatant serving as a comparative control. Highly purified T4 phages were rapidly internalized by mammalian cells and accumulated within macropinosomes but did not activate the inflammatory DNA response TLR9 or cGAS-STING pathways. Following 8 hours of incubation with T4 phage, whole cell lysates were analyzed via antibody microarray that detected expression and phosphorylation levels of human signaling proteins. T4 phage application led to the activation of AKT-dependent pathways, resulting in an increase in cell metabolism, survival, and actin reorganization, the last being critical for macropinocytosis and potentially regulating a positive feedback loop to drive further phage internalization. T4 phages additionally down-regulated CDK1 and its downstream effectors, leading to an inhibition of cell cycle progression and an increase in cellular growth through a prolonged G1 phase. These interactions demonstrate that highly purified T4 phages do not activate DNA-mediated inflammatory pathways but do trigger protein phosphorylation cascades that promote cellular growth and survival. We conclude that mammalian cells are internalizing bacteriophages as a resource to promote cellular growth and metabolism.

**Funding:** This work was funded by the Australian Research Council DECRA Fellowship (DE170100525 to JJB), National Health and Medical Research Council (NHMRC: 1156588 to JJB & 1125704 to GWM), and the Perpetual Trustees Australia award (2018HIG00007 to JJB). The funders had no role in study design, data collection and analysis, decision to publish, or preparation of the manuscript.

**Competing interests:** JJB has a patent application related to this work (WO2018129536A1).

**Abbreviations:** BMDM, bone marrow-derived macrophage; CDK, cyclin-dependent kinase; CHEK1, checkpoint kinase 1; DMSO, dimethylsulfoxide; DPBS, Dulbecco's phosphate-buffered saline; FBS, fetal bovine serum; GPCR, G protein-coupled receptor; HBSS, Hank's Balanced Salt Solution; HyD, hybrid detector; IRS-1, insulin receptor substrate 1; KO, knock-out; LB, lysogeny broth; MEM, Modified Eagle Medium; PI, propidium iodide; PoT, Phage on Tap; RTK, receptor tyrosine kinase; WT, wild-type.

## Introduction

Bacteriophages, also called phages, are viruses that infect and kill bacteria, their natural hosts. Phages are ubiquitous across environments and are intrinsic components of our microbiomes, colonizing all niches of the body [1]. As such, the human body is frequently and continuously exposed to a diverse community of phages [2,3]. This is especially true within the gut, which houses a high-diversity microbial community [4]. Phages are essential components of the gut and participate in the genetic diversification and individualization of the gut microbiome throughout our life span [5–10]. While phages facilitate many changes to these gut microbial communities, they are also known to interact with the underlying mammalian cell layers [1,3,11–13]. Mammalian cells can engulf phages via a variety of mechanisms, leading to the internalization and accumulation of active phages [3,12–14]. Phages have been shown to bind specific mammalian cellular receptors, triggering receptor-mediated endocytosis [15,16]. However, the predominant mechanism by which phages have been shown to enter mammalian cells is through nonspecific internalization via macropinocytosis [1,12,13].

Macropinocytosis is an actin-based process characterized by the nonspecific internalization of extracellular fluid, nutrients, and potential microorganisms in large endocytic vesicles known as macropinosomes. Macropinocytosis is essential for cellular growth and cell proliferation as it allows the cell to access extracellular nutrients [17–21]. Additionally, macropinocytosis can sample and subsequently detect pathogens and foreign nucleic acids, leading to the activation of the innate immune system [22]. Macropinocytosis of phages by mammalian cells is a nonspecific process whereby cells create actin-mediated ruffles elongating outwards from the cytosol to engulf the extracellular milieu and any phages residing within it. Phages internalized via this pathway steadily accumulate within intracellular macropinosomes [13]. The downstream processing of the macropinosome can follow various pathways, including fusion with other endocytic vesicles, fusion with lysosomes leading to acidification and the inactivation of internalized components, recycling and transport to plasma membranes, and constitutive exocytosis [20].

Once inside the cell, phages may stimulate a diverse array of effects. The few studies that have investigated the cellular and innate immune response to phages have hinted at 2 opposing responses. On the one hand, certain phages like T4 or T7 are known to induce anti-inflammatory responses [23–27]. On the other, a growing number of studies have shown pro-inflammatory immune responses and inflammation to specific phages [28–30], including the filamentous phage Pf [29]. As such, it appears that certain phages may induce anti- or pro-inflammatory responses, highlighting an underlying specificity with the cellular detection of specific viral types. It remains mechanistically unclear how phages interact with and modulate the mammalian cells' innate immune response and how these interactions can influence downstream cellular processes.

In this study, we investigate whether phage T4 can modulate cellular and innate immune pathways across in vitro cell lines. We demonstrate that phages were internalized by mammalian cells via macropinocytosis, with functional phages continually accumulating within macropinosomes [12,13]. All phage preparations were highly purified and confirmed to be free of bacterial endotoxins [31,32]. To further ensure the cellular responses detected were elicited by the phages themselves, we used a comparative control that consisted of the highly purified phage lysate filtered 4 times through a 0.02-μm filter to remove phage particles and obtain a phage-free lysate composed of the background supernatant. Using these samples, we performed luciferase assays, cellular proliferation assays, and interrogated antibody microarrays to probe the cellular and innate immune changes induced by the presence of T4 phage. From these results, we showed that T4 phage do not activate DNA-mediated inflammatory pathways,

and rather, that phage internalization induced broad cellular signaling cascades affecting cell metabolism, survival, cell cycle progression, and growth.

## Results

### T4 phage do not activate the intracellular DNA-sensing receptors TLR9 and cGAS-STING

We focused on bacteriophage T4, a virulent *Tevenvirinae* phage, with an approximately 200-nm long myovirus morphology and a 168,903-bp genome, which infects *Escherichia coli* [33,34]. This phage was selected as it was previously demonstrated to be internalized by mammalian cells, accumulating intracellularly within macropinosomes over time [12,13]. Phage lysates were purified and concentrated via ultrafiltration following the Phage on Tap protocol to produce a single, high-titer phage stock that was used for all subsequent assays [31,32]. Phage stocks were treated with DNase and RNase to remove extracellular nucleic acids followed by endotoxin removal using 1-Octanol washes. As bacterial endotoxins are known to trigger an innate immune response in TLR4 expressing mammalian cells, we ensured all phage samples were deplete of endotoxin (<1 EU/mL). Despite this, there remained the possibility of bacterial components (i.e., proteins, polysaccharides, nucleic acids) persisting at low levels within our phage lysates. To address this, we passed the phage lysate through a 0.02-μm filter 4 times to generate a phage-free lysate that would also contain any residual bacterial components, henceforth referred to as "Filter control," which served as a comparative control to ensure cellular responses were phage driven and not induced by any bacterial residues or buffer contaminants. We further prepared 2 additional samples, referred to as "Capsid-only" and "Phage DNA." The Capsid-only sample was prepared by successive heat treatment of T4 phages to break the capsid followed by DNase treatment to eliminate the DNA and thus contains a mixture of partially degraded phage proteins. The Phage DNA sample was prepared via DNA extraction using a column to produce a phage-genome sized band when visualized on an agarose gel, with further DNA integrity checked by T4-specific PCR.

We then investigated whether T4 phages and associated controls could be internalized by our in vitro tissue culture cells, and whether they activate key intracellular nucleic acid receptors, which stimulate downstream pro-inflammatory immune pathways. T4 phages were applied to both A549 human lung epithelial cells and MDCK-I dog kidney epithelial cell lines and were visualized as distinct fluorescent puncta within the cell cytoplasm, suggestive of phage accumulation within membrane-bound vesicles (Fig 1A and 1B) [12,13]. Once internalized, T4 phage and their genomic material could be recognized by the nucleic acid receptor TLR9 [35], a transmembrane protein that resides within endocytic vesicles and preferentially binds DNA from bacteria and viruses [21]. Once activated, TLR9 leads to a downstream cascade via the MyD88 pathway, resulting in the induction of inflammatory cytokines through activation of NF-κB and other transcription factors, including IRF7, which bind the IFN-β, promoter [36–39]. To test this, we used a luciferase-based luminescence assay to detect the downstream activation of TLR9 in activating the IFN-β promoter in A549 cells following the addition of either T4 phage or the Filter control [40]. We saw no activation of luciferase expression in either the T4 phage or Filter control samples, while transfection of the positive controls showed strong activation of expression from both plasmids (Figs 1C, 1D and S1). From these results, we conclude that neither NF-κB-dependent activation, nor activation by other elements that activate IFN-ß expression were induced by the internalization of T4 phages. This suggests that the T4 phage capsid remains intact and that phage DNA was not exposed nor detected by TLR9 within the macropinosome.

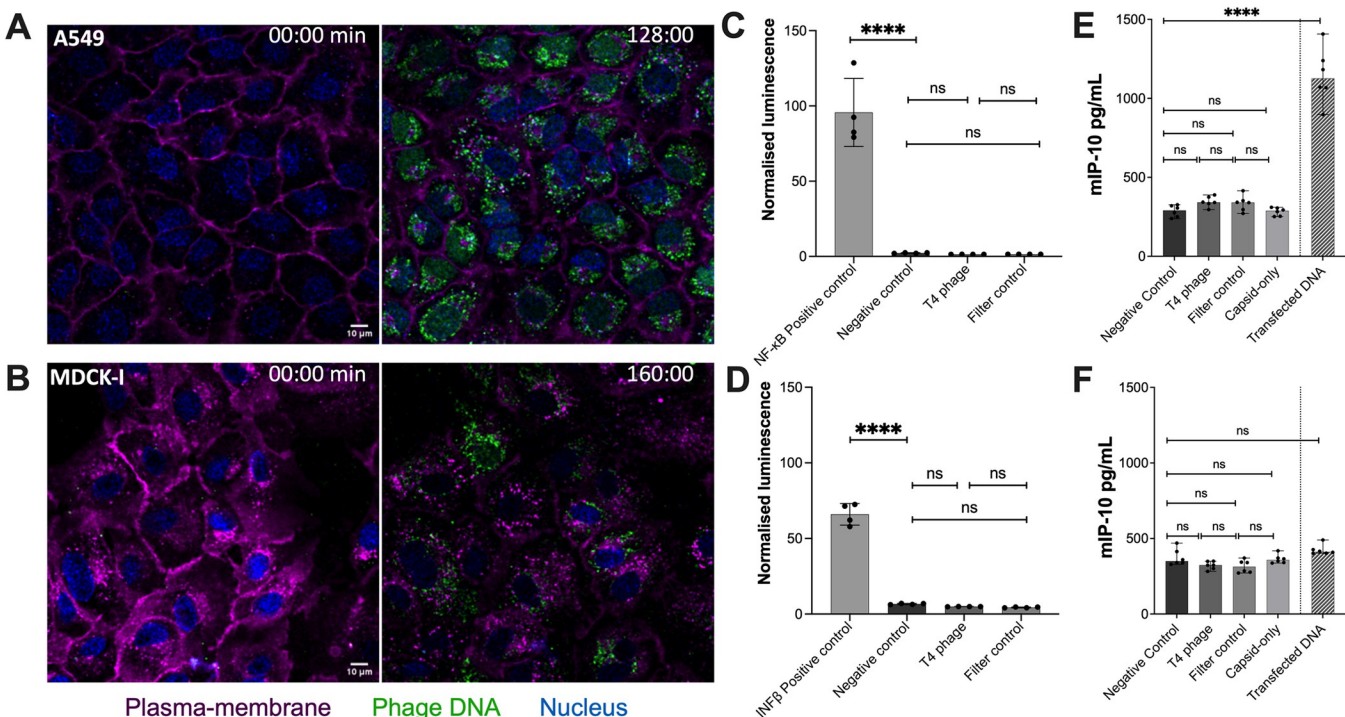

**Fig 1. Uptake of T4 phage by mammalian cells does not trigger a pro-inflammatory immune response. (A)** A549 cells and **(B)** MDCK-I cells incubated with T4 phages for 2 hours. Images were taken with a confocal microscope; the plasma membrane is shown in magenta, T4 phage DNA in green, and the cell nucleus in blue. **(C)** A549 cells transfected with NF-κB-dependent luciferase reporter plasmid, or **(D)** IFN-β promoter-dependent luciferase reporter plasmid, followed by 48 hours incubation with $10^9$ T4 phages/mL or a Filter control. Differentiated WT **(E)** or STING KO **(F)** BMDM cells were incubated for 18 hours $10^7$ T4 phages/mL, Filter control, Capsid-only or transfected with phage DNA using Lipofectamine 2000. Raw data can be found in S1 Data; each set of data follows the normality law; $P$ values between the different groups were calculated from a one-way ANOVA with multiple comparisons, shown as stars ($P < 0.0001 = $ ****; A: $F_{(3, 12)} = 31.06$; B: $F_{(3, 12)} = 2.812$; C: $F_{(4, 25)} = 5.7$; D: $F_{(4, 25)} = 0.8181$).

Following internalization and trafficking, phage particles or DNA may escape the macropinosome and gain access to the cytosol. Here, the presence of unencapsulated phage DNA would be recognized by the cGAS-STING pathway, leading to the production of IFN-β and inflammatory cytokines [41–44]. To test this, we incubated wild-type (WT) bone marrow-derived macrophages (BMDMs) or STING knock-out (KO) BMDM with either $10^7$ T4 phages/mL, Filter control, or Capsid-only samples for 18 hours. After incubation, the activation of the cGAS-STING pathway was measured by ELISA to measure IFN-β levels (Fig 1E and 1F). We saw no STING induction in either the Phages, Filter control, or Capsid-only samples. As an additional a positive control, we transfected cells with extracted T4 phage DNA using Lipofectamine 2000 to demonstrate that phage DNA can activate cGAS-STING, with phage DNA showing a strong activation of STING in the WT cells (Fig 1E). Comparatively, in the STING-KO BMDM cells, we did not see any activation of the cGAS-STING pathway for any of the controls or samples (Fig 1F). Importantly, both WT and STING-KO BMDM cells also have a functional TLR9, suggesting that the fact our STING KO cell lines did not respond to phage stimulation provides further confirmation that TLR9 is indeed not in play. In summary, highly purified T4 phage were internalized by mammalian cells but did not activate pathways downstream of TLR9, including NF-κB-dependent pathways, nor cGAS-STING signaling pathways. This suggests that internalized T4 phages capsids remain intact or are trafficked in such a way as to prevent phage DNA from being exposed and triggering the innate immune system.

## T4 phage induces protein expression and phosphorylation changes in cell signaling pathways

We utilized an antibody microarray to investigate broader cellular changes in the expression and phosphorylation of key cell signaling proteins in response to phage T4 compared with the Filter control. We used 2 protein microarrays from Kinexus Biotech, the KAM-1325 microarray that contains 1,325 pan- and phosphosite-specific antibodies covering all the main cellular signaling pathways, and the KAM-2000 microarray with 2,000 pan- and phosphosite-specific antibodies. Importantly, the MDCK-I samples were analyzed by the KAM-1325 antibody array, while the A549 samples were analyzed using the improved KAM-2000 antibody array, which includes most of the antibodies from the prior array, along with 675 additional antibodies for improved detection of cellular changes.

For deconvolution of the microarray datasets, we used the previously developed methodology by Adderley and colleagues [45]. Datasets were mapped onto a network followed by a pathway analysis, which utilizes random walks to identify chains of phosphorylation events occurring more or less frequently than expected [45]. Rather than focusing solely on the largest fold changes, this analysis identifies cellular pathway interactions to provide an interpretation of the most important pathways that are influenced by exposure to phages This provided us with an overview of the main cellular pathways that were influenced by the presence of phages (Fig 2A: MDCK-I and Fig 2B: A549; S2 and S3 Figs). From this analysis, we found 52 hits for the MDCK-I cells, and 150 hits for the A549 cells, which utilized the improved KAM-2000 antibody array. Based on this analysis, we focused our attention on 2 pathways—AKT and CDK1. These pathways were selected as they were common across the 2 antibody microarray datasets and were both associated with long network associations.

## T4 phage activates the AKT pathway promoting cell growth, survival, and macropinocytosis

The AKT signaling pathway regulates a myriad of cellular functions, including promoting cell growth, proliferation, survival, and metabolism [46]. AKT itself is a serine/threonine-specific protein kinase that is activated through extracellular growth factors, such as insulin, which are detected through receptor tyrosine kinases (RTKs) or G protein-coupled receptors (GPCRs). These receptors recruit PI3K (also called PIK3CA) to the membrane, leading to the subsequent recruitment of PDK1 (also called PDPK1), which, once activated, will phosphorylate AKT on the tyrosine amino acid, T308. Alternatively, PDK1 may recruit mTORC2, which itself will activate AKT through the S473 phosphorylation site [46,47]. Once activated, AKT and its downstream effectors will induce a broad range of cellular responses, including activation of glycolysis, protein synthesis, cell survival and proliferation, glycogen synthesis, fatty acid synthesis, and the inhibition of autophagy.

In our MDCK-I datasets, we observed the activation of AKT by both PDK1 through the S473, and SRC through the Y326 phosphorylation sites in the presence of T4 phage (Fig 2A). Once activated, AKT led to increased inhibition of BAD (also called BCL2) through the S75 phosphorylation site, which is an agonist of cell death that prevents apoptosis and enhances cell survival [47–49]. Interestingly, in our A549 dataset (Figs 2B and S3), we observed similar activation of AKT through the S473 phosphorylation site indirectly by PDK1, but also via MAPK14, known as p38α MAPK. P38α MAPK is activated through environmental stressors and pro-inflammatory cytokines and usually results in increased cell survival [50]. Further, within the A549 cells, the activation of AKT induced the phosphorylation of EZR (Ezrin) through the T567 phosphorylation site [51,52]. Importantly, EZR acts as an intermediary between the plasma membrane and the actin cytoskeleton of the cell, with its activation being required for the formation of membrane ruffles during macropinocytosis, fusion of cell-to-cell membranes, and the formation of endosomes [53].

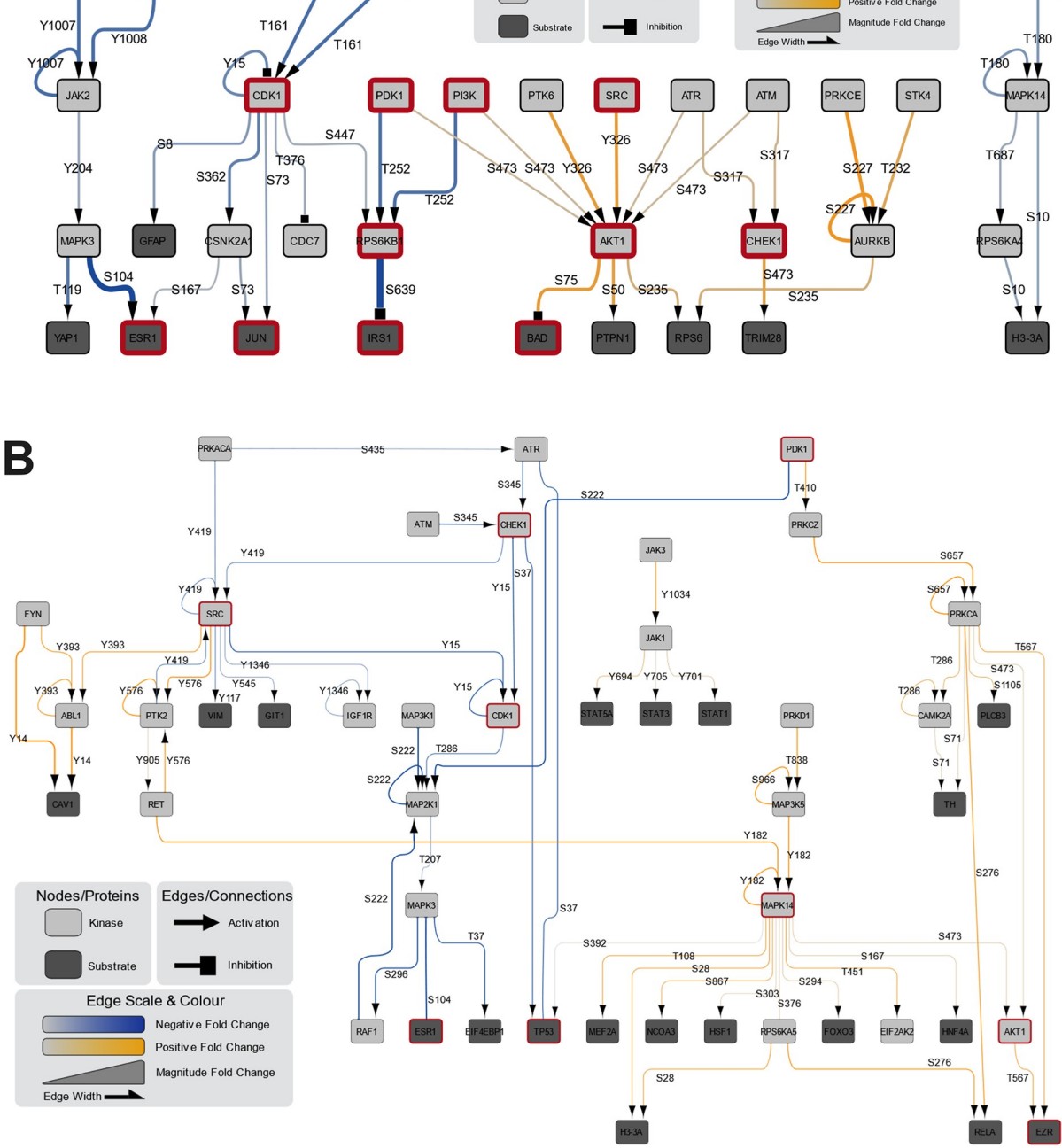

**Fig 2. Network analysis of mammalian cells treated with T4 phages. (A)** Kinexus KAM-1325 antibody microarray with MDCK-I cells after 8 hours of incubation with T4 phages. **(B)** Kinexus KAM-2000 antibody microarray with A549 cells after 8 hours of incubation with T4 phages. Figures report major cellular pathways of the main up- and down-regulated leads from the network analysis. Boxes highlighted in red are proteins discussed in this manuscript. The color gradient and arrow width indicate the $\text{Log}_2$ fold change values.

## T4 phage inhibits the CDK1 pathway to delay cell cycle progression and prolong cellular growth

The activity of cyclin-dependent kinases (CDKs) controls all aspects of cell division. CDK1 is implicated in many, if not all, cell cycle regulatory pathways and is the central hub for

regulating cells progressing through the G2 growth proliferation and mitosis phases of the cell cycle [54–56]. CDK1 is essential and sufficient to drive the mammalian cell cycle, including the entry and exit of mitosis and signaling the start of the growth proliferation phase [55,57,58].

In our MDCK-I samples (Figs 2A and S2), we observed an indirect down-regulation of CDK1 at the T161 and Y15 phosphorylation sites. The down-regulation of CDK1 inhibits cells from progressing through the G2 and mitotic phases of the cell cycle. This resulted in cascading down-regulation of other cell cycle effectors, including JUN, which is a transcription factor implicated in the prevention of apoptosis and is responsible for the progression of the cell cycle through the G1 growth phase, via down-regulation at the phosphorylation site S73 [59]. With the reduction in activation of JUN, the progression through the G1 phase of the cell cycle would be delayed, keeping cells in a prolonged state of cellular growth. Simultaneously, we observed the down-regulation in the activation of ribosomal protein p70S6K (S6 kinase beta-1 also called RPS6KB1), which regulates both cell death and proliferation [60]. This down-regulation was mediated through 2 distinct phosphorylation sites, being the T252 phospho-site, which was acted upon by PI3K and PDK1, and S447, which was acted upon by CDK1. Once inhibited, the RPS6KB1-mediated inhibition of IRS-1 (insulin receptor substrate 1) was removed. Previous reports suggest that IRS-1 can further activate PI3K, thereby leading to a positive feedback loop where IRS-1 activates PI3K to further increase AKT activation again [46]. Interestingly, we also observed the down-regulated activation of ESR1 (estrogen receptor α) in both MDCK-I and A549 arrays from the upstream effectors CDK1 and SRC [61],

In the A549 sample (Figs 2B and S3), we observed similar down-regulation of CDK1 as in the MDCK-I array, but here through the in-direct phosphorylation of Y15 site by both SRC and CHEK1 (checkpoint kinase 1). CHEK1 plays an essential role in cell cycle regulation and DNA damage response [62]. CHEK1 further regulates the G1/S transition (along with other cell cycle checkpoints) and is responsible for preventing cells with DNA damage from progressing through the cell cycle [63]. At the same time, we saw that CHEK1 was down-regulating the phosphorylation of the tumor suppressor protein TP53 through the S37 phospho-site [64,65].

## Treatment with phage leads to prolonged cell cycle growth phase and increased cellular proliferation

From the microarrays, we observed a common pattern where T4 phages induced cell cycle arrest at the G1 phase and increased cellular metabolism and growth. To validate these responses, we applied T4 phage to A549 cells to investigate differences in cell cycle and proliferation. First, we utilized a comprehensive FACS assay that measured the DNA concentration within each cell to assign them a cell cycle phase, with A549 cells being treated with either T4 phages or the Filter control for 8 or 24 hours before FACS analysis (Fig 3A) [66]. We saw no significant differences in cell cycle between the T4 phage treated and Filter control cells at the 8-hour time point. However, we did observe a significant increase in the proportion of T4 phage-treated cells in the G0/G1 phase of the cell cycle compared with the Filter control at the 24-hour time point ($P = 0.0463$). This suggests that, in line with our microarray observations, T4 phage application to A549 cells leads to a prolonged G0/G1 phase that would facilitate broad changes in metabolism, cellular growth, and cell survival.

Next, we performed an MTT cellular proliferation assay, which uses a colorimetric reduction of tetrazolium salt to measure metabolically active cells, as an indicator of cellular proliferation. A549 cells were seeded at a low density ($2 \times 10^4$ cells/mL) followed by treatment with either T4 phages, the Filter control, or a cell only control (Fig 3B). We observed a significant

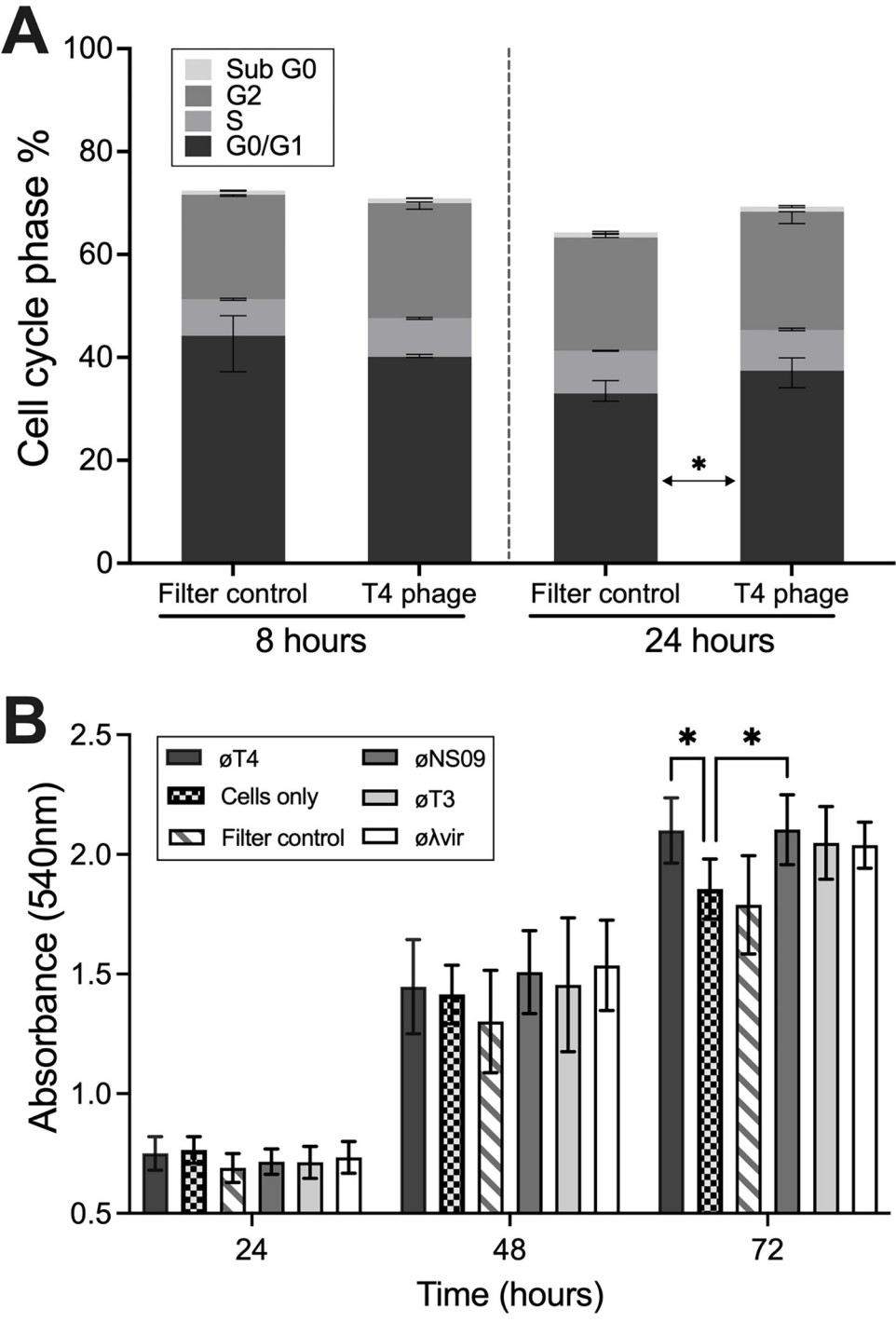

**Fig 3. Phage application to in vitro mammalian cells leads to enhanced growth and proliferation. (A)** Cell cycle stage repartition within the A549 cell population after 8- or 24-hour incubation with phages or Filter control (data are mean with error bars representing 95% CI, $n$ = 3 independent replicates with 100,000 cells analyzed). Not all the cells are included in a cell cycle phase (S4 and S5 Figs). $P$ values of each cell cycle stage between the Filter control and T4 phage were calculated using a two-way ANOVA, shown as stars ($F_{(3, 32)}$ = 2.237). **(B)** Cell proliferation assay as measured via absorbance (540 nm) using a modified MTT colorimetric assay with A549 cells incubated with phages for 24, 48, or 72 hours (data are mean with error bars representing 95% CI, $n$ = 3 independent replicates). Raw data can be found in S2 Data; $P$ values were calculated using a two-way ANOVA, shown as stars ($F_{(2, 224)}$ = 1,015).

increase in cell proliferation for the T4 phage-treated samples at 72 hours compared with cell only control ($P = 0.032$) and the Filter control ($P = 0.0044$). This suggests that T4 phage application to mammalian in vitro cells leads to enhanced cellular proliferation over a period of days. To extend these findings, we tested an additional 3 phages for their capacity to enhance cellular growth. These include øNS09, an uncharacterized virulent podovirus with a 44-kb genome, phage T3, a virulent podovirus with a 38-kb genome, and Lambda vir, a virulent siphovirus with a 48-kb genome, all of which infect the bacterial host *E. coli*. While all phages showed a trend for increased cell proliferation compared to the cell only control, only øNS09 was significantly different ($P = 0.0289$) (Fig 3B). These results suggest that addition of diverse phage to mammalian cells in vitro leads to cascade of cellular responses that promote cellular growth and proliferation.

## Discussion

We observed substantial cellular responses following the application of T4 phages to in vitro cell lines (Fig 4). Importantly, we did not observe gene activation from intracellular DNA-sensing receptors TLR9 and cGAS-STING, suggesting that internalized phages are tightly trafficked to prevent the triggering of the innate immune system. Antibody microarrays demonstrate T4 phage treatment led to broad protein phosphorylation responses that revealed common patterns across 2 cell lines. We utilized a network pathway analysis to reveal the main cellular pathways that were influenced by the presence of T4 phage [45]. This analysis suggests that exogenous T4 phages were sensed by cellular receptors (RTK and GPCRs) and internalized by nonspecific macropinocytosis. Phages promoted signaling for cell survival, proliferation, and metabolism through the activation of the AKT pathway and its downstream effectors. This is consistent with a reorganization of the actin cytoskeleton, which is critical for macropinocytosis and suggestive of a positive feedback loop stimulating further phage uptake. We experimentally validate that phages affected cell cycle progression through a prolonged G1 phase of the cell cycle, resulting in increased cellular proliferation over a period of days.

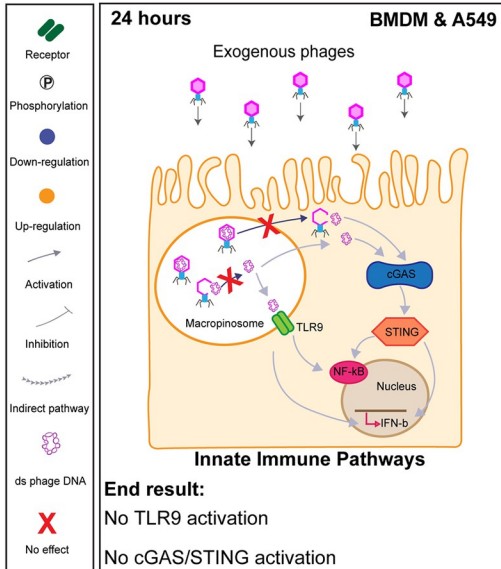
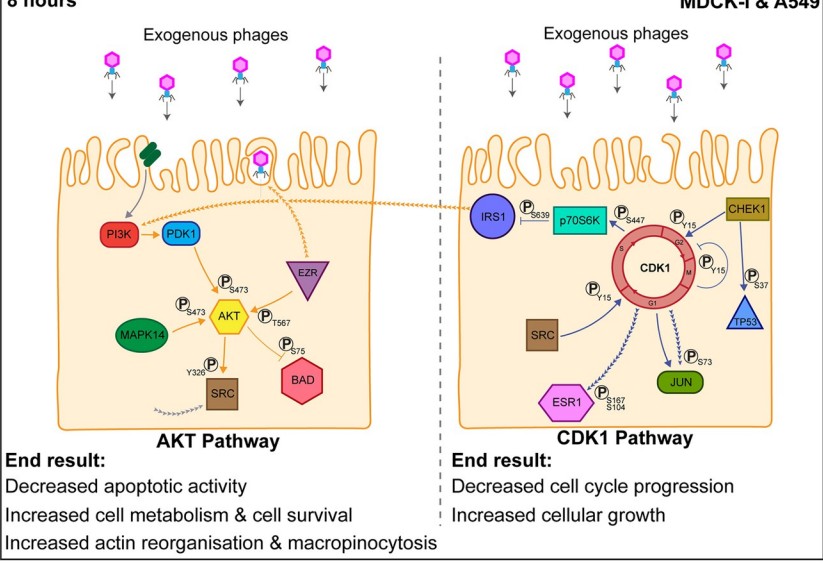

**Fig 4. Overview of the effect of exogenous phages on cellular pathways. (A)** Innate immune pathways in BMDM and A549 cells. Phage DNA is protected by the phage capsid and is not detected by the TLR9 or cGAS-STING. **(B)** The effect of phages on MDCK-I and A549 cells after 8 hours. The AKT pathway on the left and the CDK1 pathway on the right show the major cellular changes detected in response to T4 phage.

Overall, these changes suggest that while T4 phages have a benign innate immunological effect on the cell, they do broadly affect cellular response via protein phosphorylation networks. We propose that these in vitro mammalian cells are internalizing phages as a resource to maximize their growth and metabolism.

To activate key inflammatory DNA-sensing innate immune pathways, phage DNA must be accessible in either the cytosol or macropinosomes. We observed that this was rarely the case in mammalian cells treated with T4 phage. Our previous research demonstrated that macropinosome-internalized phages were maintained and accumulated within the cell over time [13]. A smaller subset of these internalized macropinosome-bound phages were capable of translocation through the basolateral side of the cell, while others colocalized with lysosomes for degradation [12]. Here, we demonstrate that internalized phage did not activate TLR9 nor the cGAS-STING pathways, both of which detect dsDNA within the macropinosome and cytosol, respectively, as demonstrated through the lack of downstream activation of NF-κB and IFN-β. This suggests 2 mechanisms: Firstly, that phage uptake and trafficking by the mammalian cell are tightly regulated by the cell with phage virions unlikely to be exposed within the cell cytoplasm, and secondly, that phages were not actively degraded nor their ejection apparatus triggered within the macropinosome. Further experiments are required to explore whether different conditions, such as incubation time, pH, temperature, or inflammation state, as well as differences between phages applied or cell lines used affect the transport and degradation of phages and the subsequent activation of innate immune response and cytokine production [28–30,67].

We utilized an antibody microarray and a pathway analysis to identify chains of protein phosphorylation events and synergistic interaction networks as an approach to decipher cellular pathways of interest [45]. From this analysis, we identified 2 main pathways—AKT and CDK1—that were affected by T4 phages across our 2 in vitro cell lines. AKT is at the center of a multitude of different cellular processes ranging from the cell cycle, apoptosis, cell survival, glucose metabolism, and the immune system [46,68]. The AKT pathway autoregulates depending on environmental stress factors, especially in response to the level of extracellular nutrients available to the cell. The activation of AKT at phosphorylation site S473 is known to activate the uptake of glucose for energy production and to promote cellular growth [69], inhibit FoxO proteins to promote anti-apoptotic and cell survival pathways [70,71], and lead to the downstream activation of the Wnt pathway that triggers the entry of cells from G0 into G1 phase [72]. Similarly, recent work investigating LNCaP epithelial cells incubated with either T4 phage or M13 filamentous phage found phages mediated the up-regulation of the PI3K/AKT pathway, leading to changes in integrin expression as well as increased cell survival [73]. We further observed that CDK1 was down-regulated by the presence of T4 phages in both cell lines. CDK1 is known to be at the center of all the control checkpoints for the cell cycle and its activation is required for cells to move between cell cycle phases [55]. The inhibition of CDK1 led to cascading down-regulation of the cell cycle effector JUN, whose activation is required for the progression of the cell cycle through the G1 phase [59]. Simultaneously, we saw a down-regulation of apoptosis via AKT inhibition of BAD and the down-regulation of TP53 phosphorylation [55,56,74]. Finally, we further observed AKT activation leading to the up-regulation of Ezrin (EZR) at phosphosite T567. Ezrin is a membrane-cytoskeleton linker that is mainly expressed in epithelial cells whose activation is required to generate membrane ruffles for macropinocytosis and for the efficient fusion of vesicles with lysosomes [53,75]. As such, the activation of Ezrin through AKT may lead to a positive feedback loop resulting in enhanced phage uptake through the macropinocytosis pathway.

Macropinocytosis is broadly utilized by mammalian cells to uptake nutrients from the extracellular space. From a macromolecular viewpoint, phages are condensed packets of

nucleotides wrapped in an amino acid shell. Phage T4 has a capsid mass of 194 MDa, 55% of which comprises DNA with a GC content of 34%, while the remaining content consists of capsid proteins containing all essential amino acids [76–78]. Both nucleotides and amino acids are essential nutrients for the growth and proliferation of mammalian cells in vitro [79,80]. Endogenous nucleotide production in particular is quite energy demanding yet is often required as the extracellular availability of nucleotides is usually negligible [81]. However, the presence of exogenous phages provides the cells with an abundant source of nucleotides accessible via macropinocytosis. Following application of T4 phage, we observed activation of macropinocytosis through PI3K, Ezrin, and growth factors [81], and the down-regulation of CHEK1, thereby preventing progression through the G1 phase of the cell cycle [63–65]. This led to the effective arrest of cell cycle progression and accumulating cells at the G1/S phase, as experimentally demonstrated through FACS analysis (Fig 3A). We further demonstrated that addition of T4 phage, along with a novel phage NS09, led to a significant increase in cell proliferation (Fig 3B). Our preliminary findings suggest that mammalian cells are internalizing phage virions, which leads to further phage uptake via macropinocytosis, with internalized phages being used as a resource to enhance cellular growth.

We propose a model whereby phages first encounter the mammalian cell membrane through diffusive mass transfer resulting in direct phage–cell interactions [12,82]. This facilitates nonspecific uptake of cell membrane-associated phages via macropinocytosis, leading to the accumulation of active virions within macropinosomes. Internalized phages steadily accumulate and remain functional within the cell for hours to days, being trafficked through diverse endosomal pathways, including vesicles cycling through macropinosomes, fusion with lysosomes for degradation, and exocytosis across the basolateral cell membrane [12,13]. While internalized T4 phages did not trigger inflammatory DNA-sensing immune pathways, they did activate expansive protein phosphorylation cascades. Broadly, these phage-mediated responses led to increased cell metabolism, cell survival, and further phage uptake via macropinocytosis, while inhibiting autophagy and cell cycle progression through the G1 phase. This was likely the result of an increased supply of phage-derived nucleotides, resulting in increased nucleotide catabolism and a prolonged stage of cellular growth. Further work is required to determine how broad these phage-mediated cellular effects might be. This should include expanding cellular and metabolomics assays on diverse cell types, particularly primary cells, rather than the cancerous cell lines utilized here, which may have a predisposition for enhanced metabolism and growth. Additional phage types and morphologies should be investigated to decipher how cellular uptake and recognition can promote the non-inflammatory, cellular growth phenotype we report, versus the inflammatory phenotype induced by certain phage species [28–30]. Open questions remain, such as why mammalian cells are internalizing phages, how certain phage species trigger aberrant cellular responses [30], whether internalized phages can infect intracellular pathogens [83], how internalized phage particles are degraded and metabolized, and mechanistically how phage-delivered nucleic acids and proteins are accessed by the cell [16,84]. This preliminary study provides novel insights into the impact phages have on mammalian systems, with broader potential implications across the fields of immunology, phage therapy, microbiome, and human health.

## Materials and methods

### Bacterial stocks and phage stocks

The bacterial strain used in this study was *E. coli B* strain HER 1024, which was cultured in lysogeny broth (LB) media (10 g tryptone, 5 g yeast extract, 10 g NaCl, in 1 L of distilled water [dH$_2$O]) at 37°C shaking overnight and used to propagate and titer T4 phages supplemented

with 10 mM $CaCl_2$ and $MgSO_4$. T4 phages were cleaned and purified using the Phage on Tap (PoT) protocol [31] and titered up to a concentration of approximately $10^{11}$ phages/mL to produce a phage stock solution that was used for all experiments. After purification, phages were treated with DNase-I and RNase and then stored in a final solution of SM Buffer (2.0 g $MgSO_4·7H_2O$, 5.8 g NaCl, 50 mL of 1 M Tris–HCl (pH 7.4), dissolved in 1 L of $dH_2O$) at 4˚C.

## Endotoxin removal

The endotoxin removal protocol followed the PoT protocol [31]. The phages lysate was cleaned 4 times with 1-Octanol to remove endotoxins from the lysate, reducing endotoxins from 5,734 EU/mL to 167 EU/mL in the final phage stock solution ($10^{11}$ phages/mL) (see also [32]). In all experiments, unless otherwise stated, phages were diluted in endotoxin-free buffers to a final concentration of $10^8$ PFU/mL (unless otherwise stated), resulting in an endotoxin concentration below 1 EU/mL.

## Control sample preparation

We prepared several control samples all using the same ultra-pure T4 phage lysate. First, in the Filter control sample, the lysate was passed 4 times through a 0.02-μm filter to remove phage particles from the lysate. The absence of phages was confirmed using a top agar assay at neat dilution with no plaques observed. Second, the capsid-only sample was obtained by breaking the phage capsid using heat treatment. Phages were heated at around 70˚C to break open the capsid and release DNA. The sample was then treated with DNase-I to degrade the phage DNA and only the empty capsids remained. Again, the absence of active phages was tested using plaque assay. Finally, the phage DNA sample was obtained using the Norgen Phage DNA isolation kit (Norgen Cat#46800) following the manufacturer's instructions and confirmed using T4 phage-specific PCR.

## Cell line stocks

The choice of A549 and MDCK-I cells was made based upon observations in our previous study demonstrating the internalization of T4 phages across both these cell lines [13]. A549 cells were grown in Ham's F-12K (Kaighn's) (also called F12-K) (Life Technologies Australia) media with 10% fetal bovine serum (FBS) (Life Technologies Australia) at 37˚C and 5% $CO_2$ and supplemented with 1% penicillin–streptomycin (Life Technologies Australia). MDCK-I cells were grown in Modified Eagle Medium (MEM) (Life Technologies Australia) with 10% FBS supplemented with 1% penicillin–streptomycin (Sigma-Aldrich, Australia) at 37˚C and 5% $CO_2$.

## Confocal microscopy

For the confocal microscopy experiment, cells were seeded in an IBIDI μ-Slide 8-well glass-bottom slide (DKSH Australia). When cells reached 80% to 90% confluency, cells were incubated for 20 minutes with the respective culture media for each cell line with 5% Hoechst 33342 stain, excitation/emission 115,361/497 nm (Life Technologies Australia) and 1% Cell-Mask deep red plasma membrane stain, excitation/emission 649/666 nm (Life Technologies Australia). After incubation cells were washed 3 times with Dulbecco's phosphate-buffered saline (DPBS) and then left in Hank's Balanced Salt Solution (HBSS) with 1% FBS until acquisition. Purified phages were labeled with 1% SYBR-Gold, excitation/emission 495/537 nm (Life Technologies Australia), following the protocol in Bichet and colleagues [32]. Around 200 μL of clean stained phages were in each well containing cells, right before the start of the acquisition (see the detailed protocol in Bichet and colleagues [32]). Cells were imaged with

HC PL APO 63×/1.40 Oil CS2 oil immersion objective by Leica SP8 confocal microscope on inverted stand with a hybrid detector (HyD) in real time. HyD detector was used in sequential mode to detect the phages. One image was acquired every 2 minutes for 2 hours. Time lapses were created through postprocessing using the FIJI software version 2.0.0-rc-68/1.52f. (See the detailed protocol in Bichet and colleagues [13].)

## Luciferase assay

A549 cells were plated at $1.5 \times 10^5$ cells/mL in 24-well cell culture plates (Corning) for 1 day. Once cells reached 70% confluency, the cells were cotransfected using Fugene HD transfecting reagent at a 1:3 ratio (Promega) along with pRL-TK *Renilla* (Renilla Luciferase, internal control; Promega) as an internal transfection control. We used the reporter plasmid pIFN-β-GL3Luc, which carriers the promoter region of IFN-β gene, and pNF-κB-Luc, which contains 5 copies of the NF-κB binding motif of the IFN-β promoter, upstream of a luciferase report gene (Firefly Luciferase) [40]. As positive controls to activate expression from the plasmids, we used FLAG-MAVS and pEF-FLAG-RIGI-I(N), which are known to activate the IFN-β promoter, through activation of NF-κB and parallel pathways, or the negative control pUC-18 (empty-vector). A transfection control well was transfected with peGFPC1 instead of pUC-18 to measure the transfection rate with a fluorescent microscope. Each well was transfected in duplicate. One day posttransfection of the reporter plasmid carrying the luciferase cassette, either T4 phage at a titer of $10^9$ PFU/mL or the Filter control were added to the transfected cells and incubated for 2 days. Cells were then incubated for 30 minutes at 4°C slowing rotating with Passive Lysis Buffer (Promega Cat#E1941). After incubation with the lysis buffers, cells were scraped and collected and spun down for 3 minutes at high speed. The supernatant was collected and kept at −20°C until analysis. The values for firefly luciferase activity were normalized to those of *Renilla* luciferase by calculating the ration of firefly to *Renilla* luminescence.

## BMDMs isolation and differentiation

BMDMs were obtained by differentiating isolated bone marrow cells from the femurs of the STING-deficient and matched WT control [85]. Briefly, bone marrow cells were flushed, washed, and differentiated in a 20% L929 cell-conditioned medium for 6 days at 37°C in a 5% $CO_2$, as described previously [86]. The use of mouse tissues was approved by the Monash University Animal Ethics Committee (MARP/2018/067).

## cGAS-STING phage assay

Approximately 24 hours before the phage treatment BDMD cells were detached by gently scraping the flasks and plated at 100,000 cells per well in a final volume of 200 μL in a 96-well plate. Around 24 hours after plating the cells, 2 μL of $10^7$ phages/mL solution, Filter control, or capsid-only samples (or 1.3 ng of phages DNA complexed with Lipofectamine 2000 as control) were added to the BMDM cells for another 18 hours. Murine IP-10 production was measured from 100 μL of the supernatant from the BMDMs using Mouse CXCL10/IP-10/CRG-2 Duo Set ELISA (R&D Systems, #Dy466) according to the manufacturer's protocol.

## KAM-1325 and KAM-2000 Kinexus antibody microarrays

Following the protocol described by Bichet and colleagues [32], confluent MDCK-I or A549 cell lines, for KAM-1325 or KAM-2000 microarrays, respectively, were incubated with T4 phages or Filter control samples for 8 hours at 37°C and 5% $CO_2$. After incubation, the cells

were scraped in lysis buffer before sonication. All samples were treated as chemically lysed proteins and followed the recommended protocol by Kinexus. The MDCK-I and A549 proteins were quantified using the Bradford protein concentration assay (Thermo Fisher Scientific). The MDCK-I samples were then incubated on the KAM-1325 array before sending the array to Kinexus for analysis, while the A549 samples were sent directly to Kinexus for analysis on the KAM-2000 array. Briefly, microarray datasets were filtered to remove low signal intensity and/or relatively high error signals compared to control signals. The network analysis was then separately run for both up- and down-regulated phosphorylation events before being assembled into a comparative pathway map [45]. Only pathways with more than 2 intermediates and with fold-changes greater than 5% CFC (% changes from control) were selected for further consideration.

## Microarray analysis

The analysis of the microarray dataset was performed using the MAPPINGS V1.0 network analysis program developed by Adderley and colleagues [45]. There was redundancy between the antibodies tested across the KAM-1325 and KAM-2000 arrays with different antibodies targeting the same protein for more precision (S2 and S3 Figs and S1 and S2 Tables). First, the signals were filtered and all signals below 1,000 units were considered as low for the KAM-1325 array and below 500 units for the KAM-2000 array and removed from the assay. Any signal with a high error relative to signal change was disregarded, and any antibody with a higher total signal error across all the arrays compared to the control arrays was disregarded. Next, each unknown substrate effect, where no known biological data were found, was considered as an activation effect for this analysis. All nodes (kinases) without a directed edge toward them were the most probable kinase for the downstream phosphorylation event. This was a consequence of the microarray data not reporting which kinase is responsible for each phosphorylation event. Independent positive and negative network analyses were analyzed, and in the case of parallel phosphorylation, only the ones with the greater magnitude values were selected and appeared on the pathway map. To ensure the proper termination of each pathway, 3 options were chosen: (1) If no other path were available after the last kinase or substrate, then the pathway was stopped. (2) If the last phosphorylation had an inhibitory effect, then the pathway was stopped as well. (3) Finally, for pathways with only one downstream option available but with no changes from the datasets between the cells incubated with the phages and the Filter control, then a percentage of fold change was assigned for each of these single paths, either between 0% and 20% for the experimental dataset and 20% for the control [45].

## FACS assay

A549 cells were plated at $8 \times 10^4$ cells/mL. Approximately 24 hours after plating the cells, phages at $10^9$ PFU/mL or Filter control samples were added to the cells and incubated for 8 or 24 hours. Cells were then washed twice with PBS; the washes were collected in a 15-mL falcon tube to prevent the loss of dead floating cells and a bias in the analysis. We added trypsin to the wells to collect the cells in the corresponding 15-mL falcon tubes. Cells were quickly centrifuged 3 minutes at 1,500$g$ before adding 1 mL of cold PBS to the pellet. While vortexing, we slowly added 2.3 mL of ice-cold 100% EtOH. Cells were incubated for 40 minutes at 4°C. Cells were centrifuged for 5 minutes at 300$g$ and resuspended in 500 μL of cold PBS. Cells were again centrifuged for 5 minutes at 300$g$ and resuspended in cell cycle buffer (in PBS, add 30 μg/mL of PI (propidium iodide), 100 μg/mL of RNase A), incubating the cells at RT in the dark for 45 minutes. The cells were centrifuged for 5 minutes at 300$g$ and resuspended in 500 μL of cold PBS before being transferred to a 5-mL round polystyrene bottom tube

(Corning). Cells were left at 4°C until FACS analysis following the [66] protocol. The experiment was performed with triplicate wells for each condition and a control well with no PI (S4 and S5 Figs). Data were generated on a 4-laser Fortessa X-20, manufactured by Becton Dickinson (BD). A total of 100,000 cells were analyzed for each assay and assigned to a cell cycle stage: G0/G1, G2, S, or Sub G0.

### Tetrazolium MTT proliferation assay

For the proliferation assay, a modified MTT (3-(4,5-dimethylthiazolyl-2)-2,5-diphenyltetrazolium bromide, Thermo Fisher Scientific) colorimetric assay was used [87,88]. A549 cells were plated at $2 \times 10^4$ cells/mL in standard 96-well plates (Thermo Fisher Scientific) and incubated at 37°C and 5% $CO_2$ overnight to recover from handling and allow attachment. The next day, 10 μL of purified phage solution at $10^{10}$ PFU/mL were added to the cells (final concentration $10^9$ PFU/mL), and plates were incubated for up to 72 hours. To account for any influence on cell proliferation potentially due to the phage lysates background, a Filter control, and a cell only control was included. At 24-, 48-, and 72-hour time points, 10 μL of 5 mg/mL MTT solution were added to all wells, including controls (final concentration 0.5 mg/mL). Cells were then incubated at 37°C and 5% $CO_2$ for 3 hours to permit intracellular formation of purple formazan crystals. Briefly, the formation of formazan is a result of tetrazolium reduction mediated by metabolically active cells and the amount that is formed broadly reflects the total number of proliferating cells. After incubation, cell culture media were removed and 50 μL of pure dimethylsulfoxide (DMSO; Merck) were added to each well to dissolve formazan crystals. To ensure complete dissolution, DMSO was mixed with cells by pipetting thoroughly, taking care to prevent bubble formation. Plates were incubated on an orbital shaker (200 rpm) in the dark for 10 minutes at 37°C. Absorbance values at 540 nm were then determined for each well using a microplate reader (Epoch2, BioTek). The assay was repeated with a total of 3 independent biological replicates with each sample tested in triplicate.

### Statistics

n represents the number of samples analyzed; each sample was performed at minimum in duplicate. The datasets were tested for normality before statistical analysis. All the statistics across this article were done using the GraphPad Prism software using one-way ANOVA or two-way ANOVA including Dunnett's multiple comparison tests. The results of the statistics are represented with stars on top of the corresponding data. All the microarray experiments were performed with only one sample; no statistics were applicable to these results.

### Supporting information

**S1 Fig. Luciferase analysis on A549 cells and T4 phages.** Individual results from the luciferase assay with IFN-ß and NF-κB from the Firefly and Renilla reporter. (**A**) Firefly IFN-ß luminescence values from each well. (**B**) Renilla IFN-ß luminescence values from each well. (**C**) IFN-ß normalized luminescence values, Firefly/Renilla, from each well. (**D**) Firefly NF-κB luminescence values from each well. (**E**) Renilla NF-κB luminescence values from each well. (**F**) NF-κB normalized luminescence values, Firefly/Renilla, from each well. Raw data can be found in S3 Data.
(TIF)

**S2 Fig. Full microarray map for the MDCK-I sample.** Showing all the interactions detected in the microarray.
(TIF)

**S3 Fig. Full microarray map for the A549 sample.** Showing all the interactions detected in the microarray.
(TIF)

**S4 Fig. FACS count results showing the distribution of the cells across the different cell cycle stages.** On the left, the 3 control samples were incubated with the Filter control for 8 hours. On the right, the 3 samples were incubated with the T4 phages for 8 hours. The cell cycle stages were set on a nonincubated sample and kept fixed for all the following analyses.
(TIF)

**S5 Fig. FACS count results showing the distribution of the cells across the different cell cycle stages.** On the left, the 3 control samples were incubated with the Filter control for 24 hours. On the right, the 3 samples were incubated with the T4 phages for 24 hours. The cell cycle stages were set on a nonincubated sample and kept fixed for all the following analyses.
(TIF)

**S1 Data. Raw numerical values for quantitative data presented in Fig 1.**
(XLSX)

**S2 Data. Raw numerical values for quantitative data presented in Fig 3.**
(XLSX)

**S3 Data. Raw numerical values for quantitative data presented in S1 Fig.**
(XLSX)

**S1 Table. Table listing the mains leads for the microarray MDCK-I sample.**
(DOCX)

**S2 Table. Table listing the mains leads for the microarray A549 sample.**
(DOCX)

## Acknowledgments

We thank the following labs for kindly providing the cell lines; Hudson Institute of Medical Research and the Oncogenic Signaling Lab for providing the A549 cell line; Stephane Chappaz for preparing the mice femurs and extracting the BMDMs. We thank Rongtuan Lin (McGill University, Canada), Naoto Ito (Gifu University Japan), Natalie Borg (RMIT, Australia), and Takashi Fujita (Kyoto University, Japan) for kindly providing the INF-ß GL3 and NF-κB, pEF-FLAG-RIGI-I(N) and FLAG-MAVS plasmids, respectively. Authors acknowledge and thank the following facilities for kindly providing equipment and guidance: Monash Micro Imaging Facility for help with microscopy acquisition, and the FlowCore for assistance with flow cytometry analysis. Marion C. Bichet was supported by Monash Graduate Scholarship (MGS).

## Author Contributions

**Conceptualization:** Marion C. Bichet, Jeremy J. Barr.

**Formal analysis:** Marion C. Bichet, Jack Adderley, Laura Avellaneda-Franco, Isabelle Magnin-Bougma, Linden J. Gearing, Ruby CY Lin, Ruzeen Patwa, Gregory W. Moseley, Christian Doerig, Jeremy J. Barr.

**Funding acquisition:** Jeremy J. Barr.

**Investigation:** Marion C. Bichet, Celine Deffrasnes, Genevieve Pepin, Ruzeen Patwa.

**Methodology:** Marion C. Bichet, Jack Adderley, Laura Avellaneda-Franco, Isabelle Magnin-Bougma, Natasha Torriero-Smith, Linden J. Gearing, Celine Deffrasnes, Cassandra David, Genevieve Pepin, Michael P. Gantier, Ruzeen Patwa, Christian Doerig.

**Project administration:** Jeremy J. Barr.

**Resources:** Michael P. Gantier, Gregory W. Moseley, Christian Doerig, Jeremy J. Barr.

**Supervision:** Jeremy J. Barr.

**Writing – original draft:** Marion C. Bichet, Jeremy J. Barr.

**Writing – review & editing:** Marion C. Bichet, Jack Adderley, Laura Avellaneda-Franco, Linden J. Gearing, Celine Deffrasnes, Cassandra David, Genevieve Pepin, Michael P. Gantier, Ruby CY Lin, Ruzeen Patwa, Gregory W. Moseley, Christian Doerig, Jeremy J. Barr.

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
