## [Editor Report · Decision Letter 0]

23 Mar 2023

Dear Dr. Barr, 

Thank you for submitting your manuscript entitled "Mammalian cells internalize bacteriophages and utilize them as a food source to enhance cellular growth and survival" for consideration as a Short Report by PLOS Biology.

Your manuscript has now been evaluated by the PLOS Biology editorial staff and I am writing to let you know that we would like to send your submission out for external peer review. 

We will consider your manuscript as a Short Report. Please select that option where it corresponds in the system when re-submitting. Short Reports present the results from a limited set of experiments that can generally be summarized in 3-4 figures or fewer. The outcomes should be self contained, rather than fitting within the narrative arc of a larger research project or article.

Once your full submission is complete, your paper will undergo a series of checks in preparation for peer review. After your manuscript has passed the checks it will be sent out for review. To provide the metadata for your submission, please Login to Editorial Manager (https://www.editorialmanager.com/pbiology) within two working days, i.e. by Mar 25 2023 11:59PM.

Kind regards,

Paula

---

Senior Editor

PLOS Biology

---

## [Decision Letter · Decision Letter 1]

23 May 2023

Dear Dr Barr,

Please allow me to first apologize for the delay in the processing of your manuscript. This delay is caused by my difficulty in recruiting reviewers for your manuscript. I am sorry for this unexpected event, and I thank you for your patience while your manuscript "Mammalian cells internalize bacteriophages and utilize them as a food source to enhance cellular growth and survival" was peer-reviewed at PLOS Biology. It has now been evaluated by the PLOS Biology editors, an Academic Editor with relevant expertise, and by several independent reviewers. 

In light of the reviews, which you will find at the end of this email, we would like to invite you to revise the work to thoroughly address the reviewers' reports.

As you will see below, the reviewers find the manuscript interesting, but find some issues that should be solved before publication, such as overstatements regarding the adaptation. Reviewer #1 thinks that it would be good to show that your results are reproducible with other phages. We consider that given that you imply that this is generalizable, it would be helpful to add at least one other phage for comparison. 

Given the extent of revision needed, we cannot make a decision about publication until we have seen the revised manuscript and your response to the reviewers' comments. Your revised manuscript is likely to be sent for further evaluation by all or a subset of the reviewers.

**IMPORTANT - SUBMITTING YOUR REVISION**

*Re-submission Checklist*

*Published Peer Review*

*PLOS Data Policy*

*Blot and Gel Data Policy*

Sincerely,

Paula

---

Senior Editor

PLOS Biology

REVIEWS:

Reviewer #1: Phage therapy.

Reviewer #2: Microbiota-host interactions.

Reviewer #1: This is a compelling Short Report manuscript, which describes a study examining internalization of phage T4 particles by cells of eukaryotic origin. The main conclusion is that phage internalization promotes growth of cells. 

The paper is generally well-written and nicely organized. The results are exciting, from the standpoint of phages promoting growth as opposed to activating innate immunity of eukaryotic cells. I do not have many comments, but one of my major concerns is that the data are reported while insinuating - but not proving - that the observations indicate cellular adaptation. Here are my major and minor comments that the authors should consider:

Major comments:

1. The use of the phrase 'food source' seems inappropriate or unnecessary; although this may capture the attention of non-biologists or the lay public, 'food' generally refers to a substance containing protein, carbohydrate, fat etc. that nourishes a cellular metabolizing organism. But when a photosynthesizing plant uses sunlight for growth, do botanists refer to this as 'food'? The paper title could just as easily read "… internalize bacteriophages as resources to enhance cellular growth …". I suggest this change to better reflect the true biology (as I understand it), and to avoid confusion by the reader. 

2. Of course, a main remaining question is WHY are the phage particles used as resources, and whether the cells are evolved via adaptation to benefit from this internalization, versus simply capitalizing on the process because the steps of internalization and downstream 'processing' of the phages are hijacked from other adaptive traits. This is not trivial to disentangle, as most purported adaptive traits are assumed rather than proven. The paper clearly hints that this is adaptive (e.g., lines 338-9: "cells are internalizing phages as a food source to maximize their growth and metabolism"). I view the data as indicating the cells capitalize on phage internalization (e.g., their nucleotides), but it seems a stretch to conclude they are tuned to use phages as food per se. I caution the authors to not overstep in their claims. Only one phage is examined, and it is unknown whether this phenomenon would work similarly for other viruses (which the authors admit). 

Relatedly, why should the cells be particularly good at internalizing and using phage T4 this way, as opposed to other phages which are perpetually or occasionally present in the microbiome? On the one hand, this is a Short Report and the authors are not expected to test widely across various phages. But on the other hand, just because this is a short paper it does not justify insinuating that this is adaptive without having looked at the phenomenon more widely. Thus, I believe the authors should be very careful in how they present the findings, so that the audience does not prematurely consider this to be adaptive. Indeed, the authors admit that other non-cancerous cells should be examined (lines 425-6). Immortalized cells are convenient, but biologically 'weird'. For this reason, studies using such cells are understandably viewed with inherent skepticism and older conclusions that ignored this weirdness are being re-examined for spurious conclusions. On the authors behalf, I would not want conclusions of the current study to be easily dismissed after publication, when viewed through a similar lens and this is my main motivation for suggesting they tread carefully. 

Minor comments:

1. Line 391. I worry that "phage T4 has most complex capsid structure of any virus" is unwarranted given our poor knowledge of the vast majority of viruses on earth. It seems better to state "of any known virus".

Reviewer #2: Mammalian cells internalize bacteriophages and utilize them as a food source to enhance cellular growth and survival.

This manuscript aimed to investigate which cellular pathways of mammalian cells are triggered following non-specific ingestion of purified phage T4 using a combination of immuno-based assays and microscopy. Bichet et al., present a strong argument for mammalian cells using bacteriophages as a source of nutrients for cellular growth. They demonstrate that phage T4 are internalised via non-specific macropinocytosis, evade triggering innate immune pathways, and instead activates growth signals (i.e., AKT-dependent pathways; protein kinase B) that are associated with, for example, increases in cell metabolism and actin reorganisation. Projects such as this may yield the next generation of biotherapeutics with novel delivery systems, and I commend the authors on tackling such a study.

In general, I have no major comments about the results, figures, and discussion sections. The overall analyses are robust, and the conclusions are contained within the limitations of this study. However, additional minor revisions are needed to strengthen the manuscript. 

Abstract:

Concise and well written. 

Introduction:

The introduction is well structured and coherent. It provides sufficient background and rationale for the remainder of the manuscript. There are some points to address that I think will help further strengthen this section. 

Line 89: "On the one hand, certain phages are known to induce…"

Please be specific in stating what these "certain" phages are. 

Line 93: "… inflammation in response to specific phages". 

Please state what these "specific" phages are. 

Line 110: Please include a concluding remark (e.g., two to three sentences) on the key findings and the implications of this study. 

Results:

The results section reads more like a narrative/thesis and some text can be either moved to the discussion section or removed entirely. 

For example, Lines 141 - 149 can be moved to the discussion section: "Once internalized, phages and their nucleic acids could be recognized by the nucleic acid receptor TLR9, a transmembrane protein that resides within endocytic vesicles and preferentially binds DNA from bacteria and viruses. TLR9 is expressed mainly in immune cells (including leukocytes and macrophages) but is also known to be expressed in a range of other cell types, including A549 cells (36). For TRL9 to be activated within the macropinosomes, phage DNA would first have to escape the phage capsid. This could happen through the triggering of the phage ejection apparatus or phage degradation due to acidic conditions found within the endosome (22). Conversely, if the phage particles stay intact, then the phage DNA would not be accessible and TLR9 should not be activated." 

Line 140: Describe instead what is observed in Figure 1. Can the authors quantitate the concentration of phage that have been internalised across the three timepoints using the images already collected? E.g., Jonkman et al., Tutorial: guidance for quantitative confocal microscopy. Nature Protocols. https://doi.org/10.1038/s41596-020-0313-9

Lines 131 - 134: Please include some basic stats of the extracted DNA. Extraction kits will fragment and may alter the form of DNA that is different to in situ. I think information on the "Phage DNA" sample would be useful (as supplementary data) for readers to know quality of DNA and size of DNA fragments. 

Line 131: Similar to above, can the authors provide information on phage proteins in the "Capsid-only" sample? Are there structural details that are lost through in vitro processing, which would otherwise trigger immune responses?

Lines 215 - 221: Please move to methods section. What other pathways were activated? 

Lines 222 - 225: "From this analysis, we found 52 hits for the MDCK-1 cells and 150 hits for the A549 cells, which utilized the improved KAM-2000 antibody array. Based on this analysis, we focused our attention on two main pathways - AKT and CDK1 - the were common across the two antibody microarrays." I would like to see a summary plot - either a Venn diagram or upset plot - highlighting the number of hits that were unique and common between the two mircoarrays; also, a more clear statement for rationale in focusing on AKT and CDK1 pathways. 

Supplementary Figures 2 and 3 depicting the network analyses is a crucial piece of data that I think should be included in the main text. 

Lines 251 - 255: Please move to discussion section. "Macropinocytosis, which is the process T4 phage utilize to access the cell, requires significant actin reorganisation to generate the membrane ruffles and cell-to-cell membrane fusion to form macropinosomes. As such, the activation of Ezrin through AKY may lead to a positive feedback loop resulting in enhanced phage uptake through the micropinocytosis pathway."

Lines 270 - 271: Please rephrase to read "With the reduction in activation of JUN, the progression through the G1 phase of the cell cycle would be delayed…"

Lines 281 - 286: Please move to discussion section. 

Lines 293 - 301: Consider moving to discussion section. 

Discussion:

Lines 425 - 426: What are some of the other limitations to this study? In vitro vs., in situ. How can the results obtained in vitro be extrapolated for use with human samples?

Line 433: Please include a concluding statement emphasising the importance of current findings. This is study represents an important step forward in better understanding the impacts of phage biology in human health - with implications in cancer biology, immunology, and the microbiome sciences. 

Methods: 

The choice of cell lines is not made clear. What was the rationale behind the choice of cell lines used in this study?

Lines 454 - 455: Missing comma. Please rephrase to read: "We prepared a new comparative control sample using ultra-pure T4 phage lysates." 

Alternatively, the copy editor at the journal should go through and correct the minor grammatical and syntax errors.

Lines 510 - 512: "The day before the phage treatment…" and "The day after 2 ul of…"

Please clarify if "the day before/after" refers to a 12 h or 24 h period, or otherwise 

Lines 518 - 532: The approach taken for the two Ab microarrays are very similar. Please merge as one subsection. 

Line 543: "…was considered as an activation effect for this analysis All nodes (kinases) without a directed..."

I think there is a period missing between "analysis" and "All"

Line 562: "The next day, phages at…". 

Please clarify the number of hours. 

Line 566: "Cells were quickly centrifuged before…". 

Please provide the speed and time of centrifugation step. 

Line 570: What does PI stand for. 

Figures: 

Figure 2: Please clarify what the error bars represent. 

Figure 3A and 3B: Under the "End Result" subheadings, replace red crosses, downward red arrows, and upward green arrows with bullet-pointed text e.g., "No TLR9 activation", "Decreased apoptotic activity", and "Increased cell metabolism & cell survival". 

The red and green arrows might confuse readers into looking for them in graphic.

---

## [Editor Report · Decision Letter 2]

14 Sep 2023

Dear Dr. Barr,

Thank you for your patience while we considered your revised manuscript "Mammalian cells internalize bacteriophages and utilize them as a resource to enhance cellular growth and survival" for publication as a Short Reports at PLOS Biology. This revised version of your manuscript has been evaluated by the PLOS Biology editors and the Academic Editor.

Based on our Academic Editor's assessment of your revision, we are likely to accept this manuscript for publication, provided you satisfactorily address the following data and other policy-related requests.

1. DATA POLICY:

A) Supplementary files (e.g., excel). Please ensure that all data files are uploaded as 'Supporting Information' and are invariably referred to (in the manuscript, figure legends, and the Description field when uploading your files) using the following format verbatim: S1 Data, S2 Data, etc. Multiple panels of a single or even several figures can be included as multiple sheets in one excel file that is saved using exactly the following convention: S1_Data.xlsx (using an underscore).

B) Deposition in a publicly available repository. Please also provide the accession code or a reviewer link so that we may view your data before publication. 

Regardless of the method selected, please ensure that you provide the individual numerical values that underlie the summary data displayed in the following figure panels as they are essential for readers to assess your analysis and to reproduce it: Figures 1CDEF, 3AB, and Supplementary Figure SF1ABCDEF.

2. We suggest a change in the title: "Mammalian cells internalize bacteriophages and use them as a resource to enhance cellular growth and survival"

We expect to receive your revised manuscript within two weeks. 

*Published Peer Review History*

*Press*

Sincerely,

Paula

---

Senior Editor,

pjaureguionieva@plos.org,

PLOS Biology

---

## [Editor Report · Decision Letter 3]

20 Sep 2023

Dear Dr Barr,

Thank you for the submission of your revised Short Reports "Mammalian cells internalize bacteriophages and use them as a resource to enhance cellular growth and survival" for publication in PLOS Biology. On behalf of my colleagues and the Academic Editor, Ken Cadwell, I am pleased to say that we can in principle accept your manuscript for publication, provided you address any remaining formatting and reporting issues. These will be detailed in an email you should receive within 2-3 business days from our colleagues in the journal operations team; no action is required from you until then. Please note that we will not be able to formally accept your manuscript and schedule it for publication until you have completed any requested changes.

PRESS

Sincerely, 

Paula Jauregui

---

Senior Editor

PLOS Biology
